# Minimal Residual Disease in Acute Myeloid Leukemia: Old and New Concepts

**DOI:** 10.3390/ijms25042150

**Published:** 2024-02-10

**Authors:** Mathias Chea, Lucie Rigolot, Alban Canali, Francois Vergez

**Affiliations:** 1Laboratoire d’Hématologie Biologique, Institut Universitaire du Cancer de Toulouse Oncopole, Centre Hospitalier Universitaire de Toulouse, 31059 Toulouse, France; mathias.chea@chu-nimes.fr (M.C.); rigolot.lucie@iuct-oncopole.fr (L.R.); canali.alban@iuct-oncopole.fr (A.C.); 2School of Medicine, Université Toulouse III Paul Sabatier, 31062 Toulouse, France

**Keywords:** MRD, AML, flow cytometry, PCR, NGS, LAIP, FlowSOM

## Abstract

Minimal residual disease (MRD) is of major importance in onco-hematology, particularly in acute myeloid leukemia (AML). MRD measures the amount of leukemia cells remaining in a patient after treatment, and is an essential tool for disease monitoring, relapse prognosis, and guiding treatment decisions. Patients with a negative MRD tend to have superior disease-free and overall survival rates. Considerable effort has been made to standardize MRD practices. A variety of techniques, including flow cytometry and molecular methods, are used to assess MRD, each with distinct strengths and weaknesses. MRD is recognized not only as a predictive biomarker, but also as a prognostic tool and marker of treatment efficacy. Expected advances in MRD assessment encompass molecular techniques such as NGS and digital PCR, as well as optimization strategies such as unsupervised flow cytometry analysis and leukemic stem cell monitoring. At present, there is no perfect method for measuring MRD, and significant advances are expected in the future to fully integrate MRD assessment into the management of AML patients.

## 1. MRD Definition

Minimal residual disease or measurable residual disease (MRD), a concept far from new, has been the subject of research articles since the early 1990s, particularly in attempts to assess residual disease in acute myeloid leukemia (AML) using murine models [1,2]. In onco-hematology, MRD is defined as the quantity of persisting residual leukemic cells in a patient after treatment, detectable with a given sensitivity (Figure 1) and exceeding the cytological sensitivity threshold (≈10^−2^) [3]. 

The assessment of MRD in chronic hematological disorders (chronic myeloid leukemia [4] and multiple myeloma [5]) as well as in acute conditions such as acute lymphoblastic leukemia [6,7] or AML is a crucial aspect enabling disease monitoring. Presently, it is widely accepted that MRD serves as a powerful and independent predictive and prognostic tool. It facilitates both the early evaluation of patients’ remission status and the likelihood of relapse, aiding in therapeutic decision-making, and serves as a biomarker of efficacy in clinical trials [4,8,9,10,11,12,13,14,15].

A meta-analysis conducted by Short et al. on 81 clinical trials spanning from January 2000 to October 2018, involving nearly 11,151 patients, revealed that patients with negative MRD exhibited a significantly superior 5-year disease-free survival (DFS) rate of 64%, in contrast to the 25% observed in patients with positive MRD. Similarly, the 5-year overall survival (OS) rate was 68% for negative MRD patients as opposed to 34% for those with positive MRD across all age groups (adult and pediatric), AML subtypes, initial sampling conditions, and times of MRD measurement [11]. Another meta-analysis conducted by Buckley et al. on 19 articles between 2005 and 2016, encompassing 1431 patients, demonstrated that pre-allograft positive MRD was correlated with decreased DFS and OS, along with a higher risk of relapse, irrespective of patient age, conditioning intensity, or MRD detection method [16].

Over several years, international efforts have aimed to standardize practices, including MRD detection methods, positive MRD thresholds, testing frequency, and biological material(s) for measurement, as well as therapeutic approaches. These efforts have yielded two consensus documents published by the European LeukemiaNet (ELN), one in 2021 [8] exclusively addressing MRD in AML (updating the 2018 document [17]) and another in 2022 focusing on the diagnosis and treatment of AML [9] (updating the 2010 and 2017 documents [18,19]). These guidelines now serve as the cornerstones on which clinicians and biologists rely for MRD assessment, guiding appropriate diagnostic and therapeutic strategies. This literature review will be structured around these two documents and recent technological advances enabling MRD measurement.

## 2. One MRD, Different Techniques 

The assessment of MRD relies, in a somewhat schematic manner, on various complementary approaches, each with its strengths and weaknesses (Figure 2). These approaches include phenotypic techniques based on flow cytometry (MFC-MRD) [20], as well as molecular methods using quantitative Polymerase Chain Reaction (qPCR-MRD) [21], and more recently, next-generation sequencing (NGS) [22] or digital PCR (dPCR) [23]. Regardless of the techniques employed, it is essential to consider the concepts of sensitivity and, more importantly, the limits of detection (LOD) defined as “the smallest measured signal that can be distinguished with a given probability from a reaction blank performed under the same conditions” and the limit of quantification (LOQ) defined as “the smallest measured value provided with an acceptable level of reliability and known uncertainty” [24]. These principles are crucial to mention with each MRD result, providing an appreciation of the limitations of each technique [9]. The multitude of techniques for MRD assessment raises questions about their concordance and, consequently, the real prognostic impact on the patient. Some studies have demonstrated that consistently negative MRD results across different techniques are associated with very favorable outcomes, consistently positive MRD is linked to reduced survival, and discordance in MRD positivity between techniques results in intermediate prognosis [25,26].

### 2.1. Phenotypic Approaches

#### 2.1.1. Multicolor Flow Cytometry (MFC)

Among the phenotypic techniques, MFC is the cornerstone upon which MRD evaluation is based and is applicable in nearly all cases [27,28,29,30,31]. The advantage of MFC over molecular techniques is its availability in most hospital centers, simplifying MRD monitoring. Currently, two complementary approaches are employed to identify residual leukemic cells (Figure 3): the “Leukemia-Associated Immunophenotype” (LAIP) approach and the “Different from Normal” (DfN) approach [8,27,32], applicable to both adult and pediatric patients [33,34]. 

#### 2.1.2. LAIP and DfN Approaches

In the LAIP approach, a specific leukemia-associated immunophenotype is determined at the patient’s diagnosis and is subsequently monitored at each MRD assessment. In the DfN approach, an aberrant leukemic phenotype is determined in comparison to healthy cells (Figure 3A,B). These abnormal phenotypes may encompass the over- or under-expression of antigens, the presence of markers from different lineages, and/or antigenic asynchrony regarding the maturation stage of the leukemic cells. Currently, ELN recommends combining both methods for optimal MFC-based MRD monitoring, recognizing that each method has its own limitations (Figure 3C,D).

With LAIP, it is crucial to use a pre-chemotherapy (medullary) sample to establish a baseline leukemic phenotype. Ideally, the LAIP should remain relatively constant over time after treatment (though this is untrue in 25% of cases where a complete change is observed [35,36]). Additionally, high expertise is required for the interpretation of true positive MRD (there is a potential for false positives if non-specific events are “hitchhiking” in the LAIP area, Figure 3E). Moreover, different leukemic subpopulations may present different LAIPs, further complicating the interpretation [32] (Figure 3G). 

To complement LAIP, the more general DfN approach allows the assessment of all populations that deviate from physiological hematopoiesis without the need for a pre-treatment sample or a stable phenotype over time (Figure 3B,D,H). However, this approach still requires high expertise and knowledge of normal and/or regenerative hematopoiesis, making inter-laboratory reproducibility more challenging [37] (Figure 3F).

#### 2.1.3. Current Technical Aspects

Currently, well-established flow cytometry protocols for assessing LAIP and DfN have been developed [32,38]. Significant international efforts have been made in recent years to establish a universal strategy to improve inter- and intra-laboratory reproducibility in MFC, exemplified by the HARMONEMIA project [39], which included nearly 17 laboratories to standardize various diagnostic panels and photomultiplier parameters. ELN proposes a harmonization of practices using a basic panel with a minimum of eight fluorochromes, including CD34, CD117, CD45, CD33, CD13, CD56, CD7, and HLA-DR. In cases where a monocytic/myelomonocytic blast component is suspected (5–10% of AML cases [20]), an additional “monocytic” tube can be added, including markers CD33, CD45, CD64, CD11b, CD14, CD4, CD34, and HLA-DR [8,9,40]. The gating strategy is based on the detection of a population of CD4+HLA-DR+CD64+ that has lost CD14 expression, or the loss of expression of HLA-DR, CD4, or CD64 on CD14+ cells. The analysis of a maximum of relevant events is based on FSC (forward scatter) versus time plots and doublet exclusion plots to optimize sensitivity. To reach the ELN-recommended MRD positivity threshold of 10^−3^ leucocytes (CD45+ cells) with the phenotype of interest, it is necessary to acquire a minimum of 500,000 to 1,000,000 viable cells, requiring a good-quality, non-diluted bone marrow sample processed within 3 days at room temperature [8].

#### 2.1.4. Strengths and Weaknesses

Among the techniques for MRD monitoring, MFC represents an accessible, cost-effective method applicable in the vast majority of AML cases (>90% [8,9,34,41]). It can provide rapid results (1–2 days) with good sensitivity (10^−3^ to 10^−4^), although currently, no therapeutic decisions are made based on routine MRD results. However, there are still technical and clinical aspects that can be improved.

On the technical side, MFC still relies on sample quality, requiring work on fresh, non-diluted bone marrow samples [20]. Poor sample quality could lead to false negatives and reduce the technique’s sensitivity. Another significant challenge is the lack of inter-laboratory reproducibility, which stems mainly from the intrinsic technical aspects of the method, such as variations in antibody clones, fluorochromes, and expertise.

Clinically, the accepted positivity threshold is ≥10^−3^ CD45+ cells with the phenotype of interest, allowing the potential background noise detected in normal or regenerating bone marrows to be overcome [42]. However, around a quarter of patients still experience relapse despite negative MRD [12], and it seems that using an undetectable MRD value as the positivity threshold with the sensitivity of current MFC techniques (i.e., up to 10^−5^ [41,43]) could be more relevant [44,45].

For instance, the GIMEMA AML 1310 study [46], conducted from January 2012 to October 2017, demonstrated the feasibility of using MRD as a decision-making tool guiding post-consolidation treatment choices in young subjects with intermediate-risk de novo AML. The study used a positivity threshold of 0.35 × 10^−4^ leukemic cells instead of 10^−3^.

### 2.2. Molecular Approaches

Molecular approaches can be more sensitive than phenotypic approaches for monitoring MRD, reaching an LOD of 10^−4^. They follow genetic abnormalities identified by molecular biology at diagnosis in leukemic cells, using PCR-based methods or NGS-based methods. Currently, qPCR methods are considered the gold standard and are the most widely used and standardized techniques for monitoring molecular MRD. The application of more recent technologies such as NGS and dPCR for molecular MRD is becoming increasingly widespread and may be able to overcome the limitations of qPCR.

According to ELN recommendations, molecular MRD should be assessed on ≥10 mL of peripheral blood or 5 mL of bone marrow from the first pull. Sampling can be conducted in EDTA or heparin tubes, although a potential inhibitory effect of heparin has been described, and should be assessed during MRD assay validation [8]. It is also recommended that the same cell isolation method (Ficoll separation or whole blood lysis) be used on samples for all MRD assessments throughout the patient’s follow-up [8].

The markers used for the evaluation of molecular MRD should be genetic alterations that are selectively expressed in leukemic cells, can be detected with high sensitivity, and exhibit relative stability at relapse [8,17,47]. 

#### Quantitative PCR (qPCR)

qPCR is the oldest method for molecular MRD, and has been used to follow *PML::RARA* fusion transcripts in acute promyelocytic leukemia since as early as the 1990s [48,49]. This technique can be applied to DNA or cDNA, with the latter being recommended for fusion transcripts and *NPM1* mutations due to their high expression levels, enabling better sensitivity. cDNA is therefore widely used for molecular MRD, requiring a reverse transcription (RT) step from RNA extracts [17]. 

Quantitative MRD data are typically obtained using real-time qPCR, which allows the quantification of PCR products during the exponential phase of the amplification process by detecting fluorescent signals during or after each PCR cycle. These signals are emitted either by non-specific fluorescent DNA intercalants (e.g., SYBR Green I dye) or by specific probes, mainly hydrolysis probes (e.g., TaqMan) or hybridization probes (e.g., LightCycler) [50]. Molecular target quantification is based on the measurement of a threshold cycle (Ct), the PCR cycle at which the fluorescence signal first rises above background noise. This Ct value is directly proportional to the amount of target sequence present in the sample, so the comparison with standards of known DNA concentration (e.g., plasmid dilutions) determines the concentration of the target in the sample. To correct for variations in RNA quality and quantity, these values are normalized to a control gene (*ABL1*) and expressed as a ratio [17,50]. According to ELN recommendations, molecular MRD by qPCR should be considered positive if Ct < 40 in at least two of three replicates [8].

Due to the widespread use of qPCR for molecular MRD, the need to standardize the technical protocols quickly arose, and in the early 2000s, recommendations were issued by several working groups [51,52], in particular, the Europe Against Cancer (EAC) program, defining optimal primer and probe designs and protocols for the real-time qPCR analysis of the main AML-associated fusion genes, and establishing *ABL1* as the most reliable internal control gene [53].

Currently, qPCR techniques can achieve sensitivities of 10^−4^ to 10^−6^ [21,23,54]. The use of qPCR is particularly well suited for MRD monitoring in AML cases where a fusion gene or a mutated transcript is highly expressed, which is true in approximately 50% of AML cases [23,54,55]. 

The main fusion genes monitored by qPCR are *RUNX1::RUNX1T1* (formerly *AML1-ETO*, resulting from t(8;21) rearrangement) and *CBFB::MYH11* (resulting from inv(16) or t(16;16)) in Core Binding Factor (CBF) AML [56], and *PML::RARA* in acute promyelocytic leukemias (APLs) [57]. Other recurrent fusion genes in AML such as *KMT2A::MLLT3* and *DEK::NUP214* are less commonly used as molecular markers for MRD because commercial standards are less available and qPCR protocols are less validated. qPCR can also be used to monitor other genetic abnormalities, mainly *NPM1* mutations [58], and, less frequently, other insertions/duplications (indels) such as *FLT3*-ITD, point mutations of *IDH1* and *IDH2*, and the gene overexpression of *WT1* or *EVI1* [59,60,61].

*NPM1* alterations are particularly suitable for MRD monitoring by qPCR as they are highly specific to AML, highly expressed, and stable at relapse [21,62]. However, it is necessary to know the type of *NPM1* mutation identified at diagnosis in order to monitor MRD, and for other mutations than types A, B, and D, qPCR protocols are less standardized and may pose technical difficulties. As it allows quantification without a calibration range, dPCR may be an appropriate alternative method for the following of those rarer *NPM1* mutations [63,64]. Although occurring in relatively rare cases, clonal evolution with the loss of *NPM1* mutation at relapse can also be a limiting factor for the monitoring of MRD [65].

qPCR methods can be used to monitor other molecular markers, such as *FLT3*-ITD and the overexpression of *WT1*, which are relatively common genetic abnormalities in AML: *FLT3*-ITD is present in 25% of adult AML cases [66], and nearly 75% of AML cases overexpress *WT1* [67]. The increasing use of targeted therapies such as gilteritinib or midostaurin (tyrosine kinase inhibitors targeting *FLT3* and other kinases) in *FLT3*-mutated AML makes the monitoring of this marker even more enticing. However, *FLT3* mutations are highly unstable from diagnosis to relapse, making monitoring particularly challenging, and because *FLT3*-ITD varies in size and location from patient to patient, it is necessary to create a qPCR technique for each patient that cannot be standardized and therefore lacks sensitivity. NGS could be an alternative method for *FLT3*-ITD MRD, but it requires specific and complex bioinformatic algorithms as ITDs are difficult to detect using classical pipelines [68,69]. For *WT1*, the physiological expression of this gene in healthy hematopoietic cells is problematic because it generates a lot of background noise. Therefore, the sensitivity and specificity of *WT1* overexpression is too low to recommend the use of this marker for MRD monitoring, unless there are no other markers available for the patient, including cytometric markers [8,17].

## 3. Current Knowledge about MRD in AML

### 3.1. Clinical Trials and Clinical Role 

The current recommendations do not yet advocate the use of MRD to adjust patient treatment in clinical practice [8,9]. Nevertheless, it now appears well established that MRD plays a role as a predictive biomarker for relapse risk (1), a relevant prognostic tool for assessing overall or progression-free survival (2), a marker of treatment effectiveness (3), and, more recently, as a surrogate endpoint (4) in clinical trials to accelerate the evaluation and availability of new therapeutic strategies [70].
(1)As a predictive biomarker for relapse risk, The HOVON/SAKK AML 42A study [30] involving 517 AML patients under 60 demonstrated that MRD was an independent prognostic factor, distinguishing high-risk relapse patients from those with better overall survival. This was later confirmed by the meta-analysis by Short et al. [11]. Yuan et al. [71] showed that pre-transplant MRD positivity, high white blood cell counts, resistance to chemotherapy, or a positive DNMT3A mutation status were associated with an increased risk of relapse.(2)As a prognostic tool for survival, Zhang et al. [72] confirmed, through a retrospective analysis of three clinical trials, that positive MRD is consistently associated with a bleak prognosis regardless of the ELN risk group. Specifically, favorable or intermediate-risk patients with positive MRD after one or two cycles of chemotherapy had a higher risk of relapse and lower overall survival than patients with negative MRD.(3)As a marker of treatment effectiveness, the phase II SORMAIN trial [73] demonstrated the utility of MRD as a predictive criterion for the effectiveness of sorafenib, an FLT3 inhibitor. Patients with negative MRD pre-transplant or positive MRD post-transplant had better relapse-free survival under sorafenib compared to patients with undetectable MRD pre-transplant or detectable MRD post-transplant under placebo. The ARTEMIS trial [74], evaluating the effectiveness of Zedenoleucel (MT-401), an allogeneic multi-tumor-associated antigen-specific T cell therapy, also showed promising results in terms of efficacy and tolerance by converting post-allograft positive MRD to negative.(4)As a surrogate endpoint in clinical trials, using the terms “AML” and “MRD” on the ClinicalTrials.gov website, approximately 324 clinical trials have been conducted, with 143 currently recruiting patients. In 2022, ELN recognized the use of MRD as a surrogate biomarker, and Walter et al. defined the optimization criteria for clinical trials. According to Walter et al. [70], each study should be conducted as an intention-to-treat, considering every MRD-positive patient as a non-responder.

Despite these advancements, there is no consensus on a gold-standard MRD evaluation method. Multimodal assessment based on AML intrinsic characteristics, such as fusion transcript presence, may be desirable.

As the field of MRD in AML is evolving, ongoing exploration is focused on its potential as a decision-making tool for therapy. For example, the GIMEMA AML 1310 study [46] highlighted the possibility of guided therapy (allo- or autologous stem cell transplantation) based on MRD. In intermediate-risk patients, a negative MRD could potentially avoid allo-transplantation, with those exhibiting positive MRD showing better overall and disease-free survival after allo-transplantation, similar to favorable-risk patients. Another promising study by Othman et al. [75] suggests that preemptive treatment with FLT3 inhibitors during molecular relapse but hematological remission has minimal toxicity and could be beneficial for overall survival. However, conflicting results are emerging from different clinical trials, emphasizing the need for more research to assess the relevance of MRD as a tool for guided therapy. For instance, the BMT CTN 0901 trial [76] suggests that intensive myeloablative conditioning could be more effective for relapse prevention and overall survival in MRD-positive patients, whereas the FIGARO trial [77] does not show a significant improvement in survival for MRD-positive patients, even after conventional myeloablative conditioning. The ELN recommends the use of conventional myeloablative conditioning whenever possible based on patient fitness [9,78]. These findings highlight the necessity for more clinical trials to evaluate the relevance of MRD as a tool for guided therapy.

### 3.2. Current Guidelines 

The updates to the recommendations on MRD in AML by the ELN in 2021 and 2022 have laid the foundations for standardizing practices and guiding future clinical trials.

#### 3.2.1. MFC and qPCR Are the Gold Standard for MRD Monitoring

In 2021 and 2022, the ELN recognized MFC and qPCR as the two preferred techniques for monitoring MRD in routine clinical practice and/or during clinical trials. Overall, qPCR remains the gold standard when a fusion transcript is present (*NPM1*, *CBF-AML*, or *PML::RARA*). Monitoring can be conducted on bone marrow or peripheral blood at diagnosis (if >20% blasts in blood at *NPM1* and *CBF* diagnosis), after two cycles of chemotherapy, and at the end of treatment on bone marrow only. Monitoring with other fusion transcripts is also possible, but may be less robust (*KMT2A::MLLT3*, *DEK::NUP214*, *BCR::ABL1*, *WT1* expression).

For all AML cases, monitoring can be performed using MFC, but only on bone marrow at diagnosis, during follow-up, and at the end of treatment. Molecular techniques such as next-generation sequencing or digital Polymerase Chain Reaction are still under evaluation and show promising results. However, they require further assessment before they can be used as standalone methods for monitoring [79,80]. These techniques still lack standardization, and for NGS, there is an issue with mutations considered part of clonal hematopoiesis of indeterminate potential (CHIP) for NGS (e.g., *DNMT3A*, *TET2*, and *ASXL1*), which should not be used for MRD monitoring [81].

#### 3.2.2. Novel Concept of Remission and Relapse 

The ELN defines various concepts of remission, including complete remission (CR) (<5% of medullary blasts, no circulating blasts, no extramedullary blast foci, neutrophil count ≥ 1 G/L, and platelet count ≥ 100 G/L), partial hematologic remission (CRh) (CR with neutrophil count ≥ 0.5 G/L and platelet count ≥ 50 G/L), and incomplete hematologic remission (CRi) (CR with neutrophil count < 1 G/L or platelets < 100 G/L). Studies have demonstrated the prognostic and predictive value of MRD [11,82,83], and the MRD result is now incorporated into each remission concept (CRMRD-, CRhMRD-, and CRiMRD-).

As indicated in Figure 2, the positivity threshold for defining positive MRD in MFC is ≥0.1% of CD45+ cells expressing the phenotype of interest, and in qPCR, it is defined by a Ct < 40 ≥ 2 of 3 replicates. A negative MRD by qPCR is defined by a Ct ≥ 40 in ≥2 of 3 replicates. A new concept is also integrated: that of CR with detectable low-level MRD (CRMRD-LL), which refers to MRD < 2% and is equivalent to a negative MRD for *NPM1* and *CBF-AML* as it is associated with a very low risk of relapse.

The ELN also details the concepts of refractory disease, defined by the absence of CR, CRh, or CRi at the time of response evaluation, and relapsed disease by medullary blastosis ≥ 5%, blood blastosis ≥ 1% on two successive samples one week apart, or extramedullary disease. Similar to remission, MRD is also included in the definition of relapse and is characterized by the conversion from negative to positive MRD regardless of the detection method or an increase in the copy number of one log10 between two positive MRDs for patients with CRMRDLL, CRhMRDLL, or CRiMRDLL.

All of these concepts do not have absolute truth value. Indeed, some patients will relapse despite negative MRD, and others will remain in remission despite positive MRD and will be clinically considered to have negative MRD due to the low risk of relapse (CRMRDLL) [15,84,85]. 

## 4. New Perspectives

There is no single technique for evaluating MRD. qPCR has the disadvantage of relying on the presence or absence of fusion transcripts and is only applicable in half of the cases. Flow cytometry heavily depends on sample quality and the expertise of the biologist, and it is generally constrained by the number of fluorochromes that can be used in the detection panel (8 to 12 colors for most centers). Various pathways have been explored to optimize MRD detection, including improvements related to the technique itself or the MRD target.

Currently, molecular techniques such as NGS and dPCR are among the future breakthroughs being studied. Other promising optimization approaches seem to revolve around either unsupervised analysis in multiparameter flow cytometry to enhance data visualization and interpretation, or the detection of a specific target closely associated with the disease: leukemic stem cells (LSCs).

### 4.1. Next-Generation Sequencing 

Since its development in the early 2000s, NGS has been increasingly used to detect small genetic alterations, such as point mutations and small insertions or deletions, and has gradually replaced Sanger sequencing in clinical practice for the molecular characterization of AML at diagnosis. Three main NGS approaches can be used: whole genome sequencing (WGS), whole exome sequencing (WES) and targeted sequencing, which provide DNA sequencing data for the whole genome, all coding sequences (2% of the genome), and multiple genes of interest, respectively. As such, these technologies are of interest for MRD because all patient-specific mutations can be followed simultaneously in a single assay, allowing MRD monitoring in AML cases where there is no molecular marker targetable by qPCR, and can also provide information on clonal evolution by detecting the gain and/or loss of mutations, which may have implications for predicting outcomes [86,87]. The quantitative measure of the variant allele frequency (VAF) of the different mutations can help define the clonal architecture of the leukemia, as mutations with similar VAFs are presumed to be carried by the same clone, with mutations with the highest VAF on the “founding clone” and mutations with the lower VAFs on distinct subclones. MRD monitoring by NGS can therefore simultaneously track the leukemic founding clones and the different subclones that may be responsible for future relapses [86,88,89]. 

To implement an NGS-based MRD method in clinical practice, it is necessary to find a compromise between the breadth of the sequenced region, which defines the ability to exhaustively detect all clonal mutations, and the depth of sequencing required to obtain a sufficient number of “reads” (i.e., DNA sequences) at each genomic position for highly sensitive MRD assessment. Therefore, WGS and WES, which are exhaustive or near-exhaustive for clonal mutations, are limited by the low sequencing depth that can be achieved, making it difficult to achieve sensitivity appropriate for MRD monitoring (e.g., WES can reach a sensitivity of 10^−2^ at best) [89]. “Deep-sequencing” approaches for NGS-based MRD use targeted NGS, which is less exhaustive, but allows for a greater depth of coverage of sequenced regions and therefore higher sensitivity than WES and WGS (up to 10^−4^). These methods generate significant amounts of data, and, as such, require large informatic resources for storage, specific bioinformatic pipelines to process the sequenced reads (i.e., alignment to reference genome, variant calling and annotation), and high biological expertise in variant interpretation, which may not be available in all MRD laboratories, but are mandatory to significantly contribute to MRD measurement.

In recent years, numerous studies have evaluated the use of NGS for MRD monitoring [8,90,91]. The first studies in 2012 showed that NGS could be used to assess molecular MRD, and that the results were highly concordant with RQ-PCR (95% for *NPM1* mutations in a study by Thol et al.) [92]. A good correlation between NGS-based MRD and flow-cytometry-based MRD has also been demonstrated by Patkar et al. [93], with almost 80% concordance between the two techniques. Several studies have also shown that molecular MRD assessment using NGS is an independent predictor of relapse and survival [26,94,95]. For example, another study by Thol et al. [96] showed that NGS-based MRD was applicable to all 96 patients in complete remission and that it was an independent predictor of relapse risk after undergoing stem cell transplantation (HR of 5.58 for NGS-positive MRD vs. NGS-negative MRD). In another study, Heuser et al. [81] confirmed that NGS-MRD was an independent prognostic factor for relapse and post-transplant survival in a cohort of 154 patients. 

NGS-based MRD monitoring is applicable to the majority of AML cases (89% to 100%, depending on the study) [26,96] and allows the assessment of clonal evolution, but there are still challenges to be overcome for this approach to be widely used in clinical practice for MRD assessment. The most challenging limiting factor is the sensitivity of NGS assays, which is highly dependent on the sequencing depth that can be achieved with the available sequencing instruments, and which is often reduced by intrinsic sequencing errors that represent constant background noise. To overcome these issues, error-corrected sequencing (ECS) can be used by incorporating “molecular barcodes” or unique molecular identifiers (UMIs), which are oligonucleotides that are added to sample DNA during library preparation to link sequenced reads to the original DNA molecule, thereby reducing errors and increasing the sensitivity of NGS techniques [23,89,97,98]. Another challenge when monitoring molecular MRD by NGS is distinguishing between leukemia driver mutations, which are associated with relapse, and clonal hematopoiesis-associated mutations [22,79]. Indeed, as hematopoietic cells can acquire somatic mutations in the absence of hematological malignancy (i.e., clonal hematopoiesis of indeterminate potential (CHIP)) and this state of clonal hematopoiesis can be a precursor state of AML, CHIP-associated mutations such as *DNMT3A*, *TET2*, and *ASXL1* (DTA mutations) often persist at high levels at cytological remission of AML without being associated with any prognostic value or higher risk of relapse [26,66]. For instance, Jongen-Lavrencic et al. [26], in their cohort of 482 patients with AML, highlighted the persistence of mutations after induction therapy in 51.4% of cases, among which DTA mutations were more frequent, often detected with high VAFs (up to 47%), and not associated with an increased relapse risk. In contrast, non-DTA persisting mutations were typically detected with low VAFs (<2.5%) and were associated with an increased relapse risk at 4 years (55.7% vs. 34.6% for undetectable non-DTA mutations, *p* = 0.006). Therefore, the monitoring of DTA mutations by NGS does not appear to be informative, and this may also be the case for other genes commonly identified in age-related hematopoiesis, such as *SRSF2*, *IDH1*, and *IDH2,* for which NGS-based MRD has been shown to have no impact in predicting relapse in *NPM1*-mutated AML [80]. The relatively high cost of NGS methods, the need for specific bioinformatic algorithms (e.g., for FLT3-ITD MRD monitoring) [68], the lack of standardization and reproducibility, and the absence of a consensus threshold to define positive MRD (temporarily defined as VAF ≥ 0.1% according to ELN 2022) are also limiting factors for NGS to be considered as a reference technique for MRD monitoring.

### 4.2. Digital PCR 

Digital PCR is a relatively recent method, developed in the 1990s [99] and first commercialized in the mid-2000s, that allows the absolute quantification of target DNA or RNA sequences in a sample without the need for a reference standard curve to normalize the results, and with good sensitivity [54,100] ranging from 10^−4^ to 10^−6^, making it interesting for MRD monitoring in AML.

This approach is based on partitioning the sample into thousands of individual wells or droplets (depending on the technology used), so that each contains either a single target molecule or few or no targets. The PCR amplification of the mutated target sequences is then performed directly on each individual partition using fluorescent probes so that the number of mutated sequences (positive partitions) and wild-type sequences (negative partitions) can be determined, and the absolute quantification of the target mutation in the sample is estimated using a Poisson distribution model [99,101,102].

Several recent studies have evaluated the use of dPCR for MRD monitoring in AML, particularly for *NPM1* mutations, for which dPCR has shown good correlation with qPCR-MRD [103], even for rare mutation types [63], and has also been shown to be a predictive factor for relapse in AML [104]. MRD monitoring of other mutations can also be carried out using qPCR, and has been investigated in a few studies using markers such as IDH1/IDH2 [105,106] and other AML-associated mutations [107]. Despite the results showing that dPCR is a suitable method for MRD monitoring, with higher sensitivity than NGS methods (e.g., Parkin et al. demonstrated dPCR sensitivity up to a VAF of 0.002%) [108], larger studies are needed to confirm the clinical relevance of using these mutations as MRD markers. Fusion transcripts can also be used as targets for dPCR MRD with sensitivity comparable to qPCR, which has been demonstrated for *PML::RARA* monitoring in APL [109,110]. 

A major limitation of dPCR is that it is a targeted technique, with similar difficulties to qPCR in terms of applicability, such as the need to know the exact genetic abnormality to be followed, which means that specific assays need to be developed for each molecular marker to be monitored, as commercial options for less common genetic abnormalities are limited. dPCR is also more expensive than qPCR and lacks standardization as it is less widely used. Another significant limitation is the relatively high rate of false positives reported with dPCR, which can result from poor assay design or the detection of spurious amplification signals at high numbers of PCR cycles; thus, the optimization of assay design and validation are critical steps to ensure that dPCR assays can be used for MRD measurement in clinical practice [102].

Despite these limitations, dPCR is a highly sensitive method for the molecular monitoring of MRD and may be of interest for the detection of rare mutations, such as non-A, B, or D mutations of *NPM1* [111], due to the lack of required calibration standards. A strong correlation between qPCR and dPCR has already been demonstrated for rare NPM1 variants in a cohort of 99 patients [63], and multiplex techniques of dPCR could be used to cover the majority of NPM1 mutations in MRD monitoring, including rare variants, as proposed in a study by Mencia-Trinchant et al. [64].

### 4.3. Unsupervised Technique in Cytometry as a Tool for MRD Monitoring 

With the increasing demonstrated utility in numerous articles on MRD and the widespread availability of MFC for its monitoring in almost all cases of AML, it became increasingly evident to attempt technical and analytical improvements for MRD monitoring. Significant progress has been made in MFC in recent years, notably with the advent of mass cytometry, allowing the measurement of up to 120 parameters [112,113]. Although still in the experimental stage for AML [114,115] such advancements highlight the need for continuous enhancement.

In routine practice, even with 8–12 fluorochromes, classical data visualization in flow cytometry remains limited, with not all parameter combinations explored due to the intrinsic nature of commonly designed biparametric scatter plots. The number of these plots increases exponentially with the number of obtained parameters. Consequently, the need to improve data visualization methods has arisen to reduce interpretation variability.

Numerous methods have emerged, including viSNE (visualizing data using t-SNE) [116], Principal Component Analysis (PCA) for dimensionality reduction, and clustering methods such as SPADE (Spanning-tree Progression Analysis of Density-normalized Events) [117], Citrus [118], SWIFT [119], and FlowSOM (Flow-Self Organizing Maps) [120,121,122,123].

Not all of these methods are suitable for MRD monitoring, as many require an extremely long analysis time (several hours) and a large number of cells, which may be incompatible with monitoring in samples with low cellularity after induction or consolidation therapy.

Weber et al. [119] compared 18 MFC clustering methods, with FlowSOM standing out for its rapidity and sensitivity in both mass cytometry and flow cytometry, utilizing open-source R software (Bioconductor).

According to Gassen et al. [122], FlowSOM is a technique enabling both cell clustering and dimensionality reduction, simplifying the visualization of all cells based on four steps: reading the data, building a self-organizing map, constructing a minimal spanning tree, and computing a meta-cluster.

Lacombe et al. [120] demonstrated the practical application of FlowSOM in the analysis of normal bone marrow and patients with hematological disorders at diagnosis or during follow-up, using traditional cytometry analysis software like Kaluza.

Subsequently, Vial et al. [123], using molecular MRD as a comparison technique, demonstrated the feasibility of an unsupervised analysis technique like FlowSOM in MRD monitoring in AML and in the approach recommended by the European LeukemiaNet for DfN and LAIP. Analyzing data from 40 patients with 96 follow-up points, they found concordant results with molecular MRD in 80.2% of cases and discordant results in 19.8% of cases, with a positive predictive value of 86% and a negative predictive value of 67%, surpassing the values reported in the literature [30,124]. All of these data appear promising for a more efficient evaluation of MRD in the era of big data.

### 4.4. Leukemic Stem Cell: A Promising Target for MRD Tracking

In AML, leukemic cells originate from a common progenitor known as a leukemic stem cell, possessing distinctive properties of quiescence, self-renewal, and the expression of specific survival genes (i.e., *ABCB1*/*ABCC1*/*LRP* [125]) rendering them particularly resistant to standard chemotherapies [126,127] such as daunorubicin or cytarabine [128]. These LSCs are implicated in the maintenance and relapse of AML [129] and represent a therapeutic target [130] and a perspective for patient monitoring.

A major challenge in recent years has been to precisely characterize the phenotype of these AML-associated LSCs and differentiate them from healthy hematopoietic stem cells (HSCs). Methods for their detection have been developed, and their relevance for residual disease monitoring has been evaluated. Conventional bulk analysis in flow cytometry provides a global study of leukemic cells, but fails to detect LSCs. Numerous studies have focused on defining the specific phenotype of AML-associated LSCs, particularly within the CD34+CD38− cell compartment [131,132,133] and exploring potential markers such as CD133 [134], CD123 [135], and CD33 [136].

Among these markers, the European LeukemiaNet has identified three key aberrant and LSC-specific markers not present in HSCs: CD45RA, CLL-1, and CD123 [131]. Zeijlemaker et al. [38] proposed LSC detection using a single tube composed of CD34, CD38, CD45RA, CD123, CD44, and CD33, combined with a cocktail of six antibodies in the phycoerythrin (PE) channel (Clec12a, TIM-3, CD7, CD11b, CD22, and CD56). 

Currently, the ELN does not recommend routine LSC detection, but supports its exploration in clinical research to assess its potential in patient monitoring. Monitoring MRD based on LSC is associated with patient prognosis, exhibiting higher sensitivity and lower risk of false negatives [137,138,139]. Rhenen et al. [140] demonstrated that a high CD34+CD38− cell count was associated with elevated MRD, and Zeijlemaker et al. [137] demonstrated, using a combined approach of classical MRD assessment and LSC detection, that patients with elevated MRD/LSC had reduced survival (hazard ratio (HR) of 3.62 for overall survival (OS) and 5.89 for cumulative relapse incidence) compared to patients with low MRD/LSC. This trend was also observed, to a lesser extent, in patients with low MRD/high LSC or high MRD/low LSC. Consequently, they identified a subgroup of patients with a risk approaching 100% therapeutic failure. Canali et al. [141] also assessed the performance of a combined approach involving classical MRD analysis in bulk and LSC detection, coupled with unsupervised analysis using FlowSOM. In a sample of 155 patients, a subgroup characterized by negative MRD but positive LSC and an intermediate prognosis was identified. This finding supports the value of LSC detection for refining MRD monitoring and demonstrates the feasibility of a machine learning method such as FlowSOM in this context.

Beyond MRD monitoring using LSCs, these cells hold considerable promise in treating AML relapses. Ongoing trials target specific signaling pathways active in LSCs, including the promising combination of venetoclax + azacitidine or the use of BH3 mimetics in relapsed or refractory AML [133]. Hence, a frontline combination of these targeted therapies with conventional chemotherapy could be a promising solution for eradicating the disease.

### 4.5. Circulating Leukemic Cells and Microfluidics as a Novel Target and Method for MRD Assessment

Teixeira et al. [54], in their review, introduced new potential targets for use in MRD called circulating leukemic cells (CLCs). Unlike their counterparts, circulating tumor cells (CTCs), which derive from solid tumors, the detection of CLCs is still in its early stages but holds promise for estimating MRD. Currently, monitoring MRD in peripheral blood is mainly reserved for molecular techniques, primarily due to sensitivity issues with very rare circulating elements. Consequently, there has been a growing interest in developing rapid, non-invasive, cost-effective techniques for disease monitoring.

Microfluidic methods have emerged as a solution, relying on the manipulation of minute fluid quantities (10^−9^ to 10^−18^ L) circulating through narrow channels (10–100 µm) [142], enabling single-cell analysis. Significant progress has been made in detecting and separating CTCs using this type of technique, considering capture rates, purification, detection limits, and biocompatibility [143]. These advancements are also extending to CLCs. Khoo et al. [144] developed a biochip for the separation, capture, and monitoring of leukemic cells based on the size and stiffness of CLCs. Their results showed high sensitivity (10^−4^ to 10^−5^), comparable to or even surpassing classical cytometry. Their work on stiffness could overcome one of the major limitations of microfluidic techniques, which is the size of the CLCs (13–16 µm), making their separation from normal leukocytes (7–15 µm) challenging.

Another study, conducted by Jackson et al. [145], demonstrated initial results of post-transplant MRD monitoring using microfluidic techniques. Using three microfluidic devices coated with antibodies against markers expressed in AML (CD33, CD34, and CD117) and aberrant markers (CD7 and CD56) at 39 MRD points, they achieved satisfactory results with the detection of relapse signs being faster than tests based on bone marrow samples. Additionally, they showed excellent specificity for CLCs (88–99%). All of these studies contribute valuable insights for the future development of new techniques to monitor MRD. 

## 5. Conclusions

Currently, there are no flawless methods for measuring MRD, and significant advances are expected in the future to integrate MRD assessment into the management of patients with AML. 

At present, the combination of phenotypic and molecular techniques (qPCR) appears to be the most reliable means to overcome the weaknesses of each technique and allow for optimal MRD monitoring. However, some questions remain unanswered about the concordance of the results between these techniques and the prognostic value of a discrepancy between cytometric MRD and molecular MRD. Additionally, there is the problem of limited sensitivity when only phenotypic MRD can be performed (when no molecular marker can be followed by qPCR), and the question of clonal evolution that can result in cases of relapse in MRD-negative patients.

With the combination of NGS and flow cytometry techniques, virtually 100% of MRD could be monitored in patients with AML. However, NGS-based MRD is not easily implementable in laboratories because of the cost of the extensive deep sequencing needed to combine both exhaustive genomic profiling and high sensitivity, and because of the informatic resources needed to process the data. Limitations also revolve around technical aspects (sample quality for flow cytometry, specific informatic pipelines for NGS) and scientific challenges (high expertise required for flow cytometry and NGS data analysis). 

The 2022 ELN provides essential insights for upcoming clinical studies to assess the various roles that MRD evaluation can play in technical and clinical terms, aiming to standardize practices.

Several questions and unresolved issues remain as the evolving landscape of diagnostic and therapeutic approaches may necessitate a reconsideration of established benchmarks for MRD positivity. This includes determining the optimal timeframes for monitoring and assessing the impact of emerging therapies on MRD dynamics. Additionally, it is crucial to explore whether the traditional endpoints align with the therapeutic responses observed with novel treatments, and whether adjustments are needed to better reflect treatment efficacy and patient outcomes. Addressing these issues will contribute to refining MRD monitoring strategies and enhancing their clinical relevance in the context of evolving medical practices and advancements in leukemia research and treatment. 

In the era of big data, personalized medicine, and precision medicine, the emergence of MRD as a guiding element in therapeutic management through high-throughput techniques seems to be a major focus in the coming years.

## Figures and Tables

**Figure 1 ijms-25-02150-f001:**
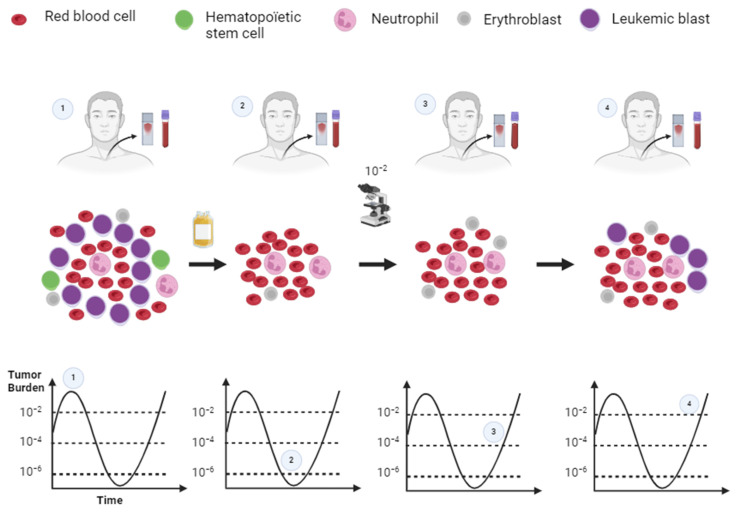
MRD principle. (**1**) AML at diagnosis. (**2**) Residual leukemic cells in cytologic remission after chemotherapy. (**3**) Relapse of AML in cytologic remission. (**4**) Cytological relapse of AML. Created with BioRender.com (accessed on 15 November 2023).

**Figure 2 ijms-25-02150-f002:**
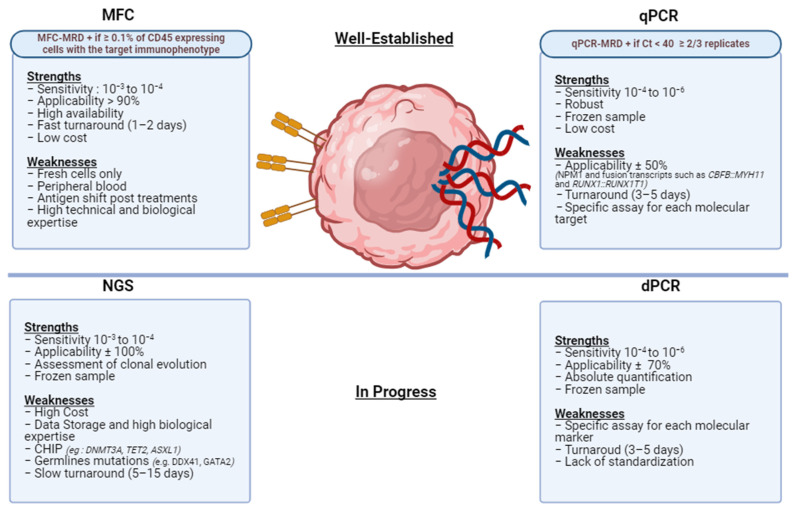
Summary of current methods for MRD monitoring. MFC: multicolor flow cytometry, qPCR: quantitative PCR, NGS: next-generation sequencing, dPCR: digital PCR, CHIP: clonal hematopoiesis of indeterminate potential. Created with BioRender.com. (accessed on 15 November 2023).

**Figure 3 ijms-25-02150-f003:**
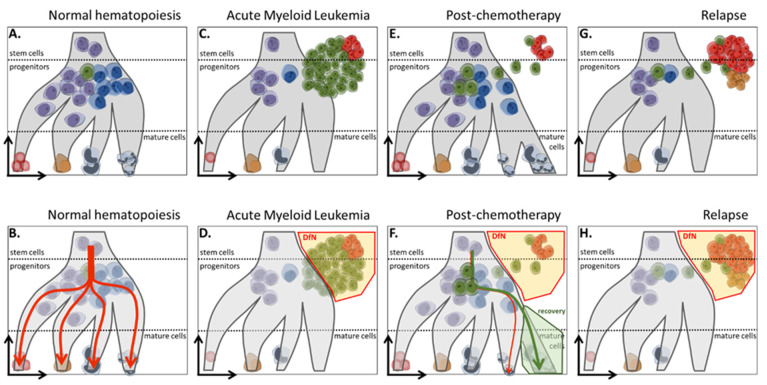
Schematic representation of MFC strategies for investigating AML MRD. Markers enable differentiation by MFC of the stem/progenitor/mature compartments of normal hematopoiesis, as well as “purple,” “blue,” and “green” phenotypic profiles (**A**). Additionally, they help determine normal differentiation pathways (red arrow, (**B**)). Leukemic blasts are distinguished by the expression of aberrant markers, which can be heterogeneous within an AML, schematized by a majority “green” LAIP and a minority “red” LAIP (**C**). The DfN strategy encompasses the portion of leukemic cells that phenotypically deviate from normal differentiation pathways (**D**). It can be observed that neither of these two strategies is perfect for the detection of MRD: (i) the “green” LAIP phenotype already exists in a subpopulation of hematopoietic progenitors, reduced under normal conditions (**A**) but increased under conditions of post-chemotherapy hematopoietic recovery (**E**), limiting the sensitivity threshold of this LAIP; (ii) the interpretation of DfN is linked to knowledge of the modifications of hematopoiesis under conditions of recovery (green arrow, (**F**)). The LAIP of relapsed blasts is modified in 90% of cases (here, relapse on a “red” LAIP profile, near-total disappearance of the “green” LAIP, and emergence of an “orange” LAIP, (**G**)). Relapse remains detectable using the DfN strategy (**H**).

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
