# Peer review of "Minimal Residual Disease in Acute Myeloid Leukemia: Old and New Concepts"

_ijms, 2024, doi:10.3390/ijms25042150_

Round 1
Reviewer 1 Report
Comments and Suggestions for Authors
This review provides a well-written and detailed overview of contemporary MRD measurement techniques. The authors have made a comprehensive effort to address various aspects of these techniques. I have a few minor suggestions that the authors may consider incorporating.
1. While Figure 2 is well-structured and designed, the use of different colors, different size boxes and positive/negative (+/-) points may not be easily understandable. Consider presenting it in a visually clearer and more easily comprehensible manner. The spelling of established are wrong and the use of scale on top does not really adds to any value.
2. Analyzing NGS data demands a specialized skill set that is not readily accessible to everyone. Consider incorporating additional details on how NGS data analysis may significantly contribute to MRD measurement.
3. A significant drawback of Digital PCR is not only the need to develop specific probe sets for each type of mutation but also the challenge of validating them. dPCR has been reported to exhibit numerous false positives. It is important to address and acknowledge this issue as well.
4. Consider rephrasing the initial sentence of the conclusion section to avoid using the term "Perfect method."
Comments on the Quality of English LanguageMinor spelling mistakes were identified. A thorough re-reading of the manuscript would assist in rectifying them.
Author Response
Dear Editors and Reviewers,
We'd like to thank you for your comments and constructive criticism and hope that our revised version will be able to respond to all your comments.
Revisions will be highlighted in blue for readability.
Reviewer 1 comment
“This review provides a well-written and detailed overview of contemporary MRD measurement techniques. The authors have made a comprehensive effort to address various aspects of these techniques. I have a few minor suggestions that the authors may consider incorporating.”
- While Figure 2 is well-structured and designed, the use of different colors, different size boxes and positive/negative (+/-) points may not be easily understandable. Consider presenting it in a visually clearer and more easily comprehensible manner. The spelling of established are wrong and the use of scale on top does not really adds to any value.
Thank you for your comment, the figure 2 with your recommendations is updated for greater clarity.
- Analyzing NGS data demands a specialized skill set that is not readily accessible to everyone. Consider incorporating additional details on how NGS data analysis may significantly contribute to MRD measurement.
This text is added (line 393-407):
To implement an NGS-based MRD method in clinical practice, it is necessary to find a compromise between the breadth of the sequenced region, which defines the ability to exhaustively detect all clonal mutations, and the depth of sequencing required to obtain a sufficient number of “reads” (i.e. DNA sequences) at each genomic position for highly sensitive MRD assessment. Therefore, WGS and WES, which are exhaustive or near-exhaustive for clonal mutations, are limited by the low sequencing depth that can be achieved, making it difficult to achieve sensitivity appropriate for MRD monitoring (e.g. WES can reach a sensitivity of 10-2 at best)[89]. “Deep-sequencing” approaches for NGS-based MRD use targeted NGS, which is less exhaustive but allows for greater depth of coverage of sequenced regions and therefore higher sensitivity than WES and WGS (up to 10-4). These methods generate significant amounts of data and as such require large informatic resources for storage, specific bioinformatic pipelines to process the sequenced reads (i.e. alignment to reference genome, variant calling and annotation), and high biological expertise in variant interpretation, which may not be available in all MRD laboratories but are mandatory to significantly contribute to MRD measurement.
- A significant drawback of Digital PCR is not only the need to develop specific probe sets for each type of mutation but also the challenge of validating them. dPCR has been reported to exhibit numerous false positives. It is important to address and acknowledge this issue as well.
This text is added (line 484-488):
Another significant limitation is the relatively high rate of false positives reported with dPCR, which can result from poor assay design or detection of spurious amplification signals at high numbers of PCR cycles, and thus optimization of assay design and validation are critical steps to ensure that dPCR assays can be used for MRD measurement in clinical practice [102].
- Consider rephrasing the initial sentence of the conclusion section to avoid using the term "Perfect method."
The sentence is rephrased as follows: “Currently, there are no flawless methods for measuring MRD”

Reviewer 2 Report
Comments and Suggestions for Authors
This is a well written review on MRD in AML and i have the following concerns.
1. The manuscript would benefit by commenting the following publication on MRD: Jongen-Lavrencic M, Grob T, N Engl J Med. 2018 Mar 29;378(13):1189-1199. The detection of persistent DTA mutations (i.e., mutations in DNMT3A, TET2, and ASXL1), which are often present in persons with age-related clonal hematopoiesis, was not correlated with an increased relapse rate. After the exclusion of persistent DTA mutations, the detection of molecular minimal residual disease was associated with a significantly higher relapse rate than no detection. Please include this reference and comment on this work.
2. Are there AML patient cases where the evaluation of MRD is not feasible with all the novel techniques and what can be done in such cases? What is the unmet need for MRD in AML in 2024 with the combination of old-classic (qPCR, flow cytometry) and all the new techniques, especially NGS?
3. Are there any gaps and unresolved problems in the MRD Evaluation - standarization in the ELN 2022 published guidelines? This is the best consensus approach until today.
Author Response
Dear Editors and Reviewers,
We'd like to thank you for your comments and constructive criticism and hope that our revised version will be able to respond to all your comments.
Revisions will be highlighted in blue for readability.
Reviewer 2 comment
“This is a well written review on MRD in AML and i have the following concerns.”
- The manuscript would benefit by commenting the following publication on MRD: Jongen-Lavrencic M, Grob T, N Engl J Med. 2018 Mar 29;378(13):1189-1199. The detection of persistent DTA mutations (i.e., mutations in DNMT3A, TET2, and ASXL1), which are often present in persons with age-related clonal hematopoiesis, was not correlated with an increased relapse rate. After the exclusion of persistent DTA mutations, the detection of molecular minimal residual disease was associated with a significantly higher relapse rate than no detection. Please include this reference and comment on this work.
This text is added (line 434-448)
Indeed, as hematopoietic cells can acquire somatic mutations in the absence of hematological malignancy (i.e. clonal hematopoiesis of indeterminate potential (CHIP)) and this state of clonal hematopoiesis can be a precursor state of AML, CHIP-associated mutations such as DNMT3A, TET2, and ASXL1 (DTA mutations) often persist at high levels at cytological remission of AML without being associated with any prognostic value or higher risk of relapse[26,66]. For instance, Jongen-Lavrencic et al [26], in their cohort of 482 patients with AML, highlight the persistence of mutations after induction therapy in 51.4% of cases, among which DTA mutations were more frequent, often detected with high VAFs (up to 47%), and not associated with an increased relapse risk. In contrast, non-DTA persisting mutations were typically detected with low VAFs (<2.5%), and were associated with an increased relapse risk at 4 years (55.7% vs. 34.6% for undetectable non-DTA mutations, p=0.006). Therefore, monitoring of DTA mutations by NGS does not appear to be informative, and this may also be the case for other genes commonly identified in age-related hematopoiesis, such as SRSF2, IDH1 and IDH2, for which NGS-based MRD has been shown to have no impact in predicting relapse in NPM1-mutated AML[80]
- Are there AML patient cases where the evaluation of MRD is not feasible with all the novel techniques and what can be done in such cases? What is the unmet need for MRD in AML in 2024 with the combination of old-classic (qPCR, flow cytometry) and all the new techniques, especially NGS?
This text is added to the conclusion:
At present, the combination of phenotypic and molecular techniques (qPCR) appears to be the most reliable means to overcome the weaknesses of each technique and allow for optimal MRD monitoring. However, some questions remain unanswered about the concordance of results between these techniques, and the prognostic value of a discrepancy between cytometric MRD and molecular MRD. Additionally, there is the problem of limited sensitivity when only phenotypic MRD can be performed (when no molecular marker can be followed by qPCR), and the question of clonal evolution that can result in cases of relapse in MRD negative patients.
With the combination of NGS and flow cytometry techniques, virtually 100% of MRD could be monitored in patients with AML. However, NGS-based MRD is not easily implementable in laboratories because of the cost of the extensive deep sequencing needed to combine both exhaustive genomic profiling and high sensitivity, and because of the informatic ressources needed to process the data. Limitations also revolve around technical aspects (sample quality for flow cytometry, specific informatic pipelines for NGS) and scientific challenges (high expertise required for flow cytometry and NGS data analysis).
- Are there any gaps and unresolved problems in the MRD Evaluation - standardization in the ELN 2022 published guidelines? This is the best consensus approach until today.
This text is added to the conclusion:
Several questions and unresolved issues remain as the evolving landscape of diagnostic and therapeutic approaches may necessitate a reconsideration of established benchmarks for MRD positivity. This includes determining the optimal timeframes for monitoring and assessing the impact of emerging therapies on MRD dynamics. Additionally, it is crucial to explore whether the traditional endpoints align with the therapeutic responses observed with novel treatments, and if adjustments are needed to better reflect treatment efficacy and patient outcomes. Addressing these issues will contribute to refining MRD monitoring strategies and enhancing their clinical relevance in the context of evolving medical practices and advancements in leukemia research and treatment.
